evolution/taxonomy and systematics/ palaeontology

Titanosauria, Gondwana, Mesozoic, phylogeny, Dinosauria, Laurasia

**Author for correspondence:**
Philip D. Mannion
e-mail: philipdmannion@gmail.com

# New information on the Cretaceous sauropod dinosaurs of Zhejiang Province, China: impact on Laurasian titanosauriform phylogeny and biogeography

Philip D. Mannion[1], Paul Upchurch[1], Xingsheng Jin[2,3] and Wenjie Zheng[2,3]

[1]Department of Earth Sciences, University College London, Gower Street, London WC1E 6BT, UK
[2]Zhejiang Museum of Natural History, Hangzhou, Zhejiang 310014, People's Republic of China
[3]State Key Laboratory of Palaeobiology and Stratigraphy (Nanjing Institute of Geology and Palaeontology, CAS), Nanjing, Jiangsu 210008, People's Republic of China

PDM, 0000-0002-9361-6941; WZ, 0000-0002-3610-806X

Titanosaurs were a globally distributed clade of Cretaceous sauropods. Historically regarded as a primarily Gondwanan radiation, there is a growing number of Eurasian taxa, with several putative titanosaurs contemporaneous with, or even pre-dating, the oldest known Southern Hemisphere remains. The early Late Cretaceous Jinhua Formation, in Zhejiang Province, China, has yielded two putative titanosaurs, *Jiangshanosaurus lixianensis* and *Dongyangosaurus sinensis*. Here, we provide a detailed re-description and diagnosis of *Jiangshanosaurus*, as well as new anatomical information on *Dongyangosaurus*. Previously, a 'derived' titanosaurian placement for *Jiangshanosaurus* was primarily based on the presence of procoelous anterior caudal centra. We show that this taxon had amphicoelous anterior-middle caudal centra. Its only titanosaurian synapomorphy is that the dorsal margins of the scapula and coracoid are approximately level with one another. *Dongyangosaurus* can clearly be differentiated from *Jiangshanosaurus*, and displays features that indicate a closer relationship to the titanosaur radiation. Revised scores for both taxa are incorporated into an expanded phylogenetic data matrix, comprising 124 taxa scored for 548 characters. Under equal weights parsimony, *Jiangshanosaurus* is recovered as a member of the non-titanosaurian East Asian somphospondylan clade Euhelopodidae, and *Dongyangosaurus* lies just outside of Titanosauria. However, when extended implied weighting is

applied, both taxa are placed within Titanosauria. Most other 'middle' Cretaceous East Asian sauropods are probably non-titanosaurian somphospondylans, but at least *Xianshanosaurus* appears to belong to the titanosaur radiation. Our analyses also recover the Early Cretaceous European sauropod *Normanniasaurus genceyi* as a 'derived' titanosaur, clustering with Gondwanan taxa. These results provide further support for a widespread diversification of titanosaurs by at least the Early Cretaceous.

## 1. Introduction

Titanosaurs were a diverse and globally distributed group of Cretaceous sauropod dinosaurs [1–3] that included the largest terrestrial animals ever known [4,5]. For much of the history of their study, titanosaurs were thought to have been a primarily Gondwanan radiation of sauropods [6–9], known mainly from the Late Cretaceous [10], with only a small number of taxa recognized from Laurasia [11–14]. In recent decades, the discovery of new titanosaurs from the latest Cretaceous of Eurasia (e.g. [15–21]), combined with the reassessment of existing taxa from East Asia [22–27], has begun to challenge this biogeographic paradigm [28]. However, the number of Gondwanan species (e.g. [29,30]) still greatly exceeds that of the Northern Hemisphere (e.g. [27,31]).

Although the vast majority of titanosaurs come from Late Cretaceous deposits [32], their fossil record extends back into the Early Cretaceous. This earlier record is best exemplified by two Gondwanan taxa from Aptian-age deposits: *Tapuiasaurus macedoi* from Brazil [32,33], and *Malawisaurus dixeyi* from Malawi [34,35], both of which preserve cranial and postcranial elements. *Triunfosaurus leonardii*, from the earliest Cretaceous (Berriasian–Early Hauterivian) of Brazil, potentially represents the stratigraphically oldest known titanosaur [36], although the fragmentary nature of this material means that its affinities should be treated with some caution [37].

Combined with their predominantly Gondwanan distribution, these Early Cretaceous remains support the view that titanosaurs probably originated in Gondwana, and probably in South America (e.g. [38]). However, there is a growing fossil record of Early–middle Cretaceous occurrences of titanosaurs from Eurasia, with a number of these specimens contemporaneous with, or even pre-dating, the oldest known Gondwanan remains [39–42]. These include: (i) *Volgatitan simbirskiensis* from the late Hauterivian of western Russia [43]; (ii) caudal vertebrae (NHMUK R5333) from the Barremian of the UK [40,44]; (iii) *Tengrisaurus starkovi* from the Barremian–Aptian of south-central Russia [45]; (iv) *Normanniasaurus genceyi* from the Albian of France [46]; (v) a caudal vertebra from the late Aptian–early Albian of Italy [47] and (vi) postcranial remains from the Cenomanian of Spain [42]. Although all of these specimens are fragmentary and highly incomplete, some appear to belong to relatively 'derived' titanosaurs; for example, *Normanniasaurus* might be an aeolosaurine [30]. Several occurrences from the 'middle' Cretaceous of East Asia might also represent titanosaurs, many of which are known from much more complete specimens, although their affinities are debated and their stratigraphic ages poorly constrained (e.g. [27,31,40,41,48–51]). These include: (i) *Daxiatitan binglingi* [52]; (ii) *Mongolosaurus haplodon* [53]; (iii) *Yongjinglong datangi* [54]; (iv) an isolated caudal vertebra described by Upchurch & Mannion [55] and reinterpreted by Whitlock *et al.* [49] (PMU 24709 [originally PMU R263]; see Poropat [56]); (v) *Xianshanosaurus shijiagouensis* [57]; (vi) *Baotianmansaurus henanensis* [58]; (vii) *Huabeisaurus allocotus* [59]; (viii) *Jiangshanosaurus lixianensis* [60] and (ix) *Dongyangosaurus sinensis* [61]. Thus, resolving the phylogenetic placements of these Eurasian taxa is critical to understanding the timing and biogeography of the early radiation of Titanosauria.

*Jiangshanosaurus lixianensis* was collected in 1977–1978 [62], and later described and named by Tang *et al.* [60], based on a partial postcranial skeleton from Jiangshan County, in the southwest of Zhejiang Province, eastern China. It was collected from the lower section of the Jinhua Formation, then dated as Albian (late Early Cretaceous), but now regarded as early Late Cretaceous in age [63]. Tang *et al.* [60] considered *Jiangshanosaurus* to be a derived titanosaur (a member of 'Titanosauridae') based on the presence of procoelous anterior caudal vertebrae. They also noted that the morphology of the pectoral girdle was most similar to that of *Alamosaurus sanjuanensis*, a saltasaurid titanosaur from the latest Cretaceous (Maastrichtian) of North America [14,64,65]. With the exception of Wilson [26], who referred *Jiangshanosaurus* to Somphospondyli based on character optimization (see also Wilson [27]), *Jiangshanosaurus* has since continued to be considered a titanosaur by all authors based on the data presented in Tang *et al.* [60]. Upchurch *et al.* [2] assigned it to Lithostrotia, and D'Emic [40] suggested possible saltasaurid affinities based on character optimization. Mannion *et al.* [41] were the

first to include *Jiangshanosaurus* in a phylogenetic analysis, in which it was recovered as a saltasaurid, in a sister taxon relationship with *Alamosaurus*. Averianov & Sues [51] subsequently argued that *Jiangshanosaurus* is unlikely to be a saltasaurid based on the absence of strong procoely in the figured caudal vertebrae [60, pl. 2], contrasting with the original written description. Consequently, Averianov & Sues [51] suggested that *Jiangshanosaurus* more probably represents a non-lithostrotian titanosaur. This latter placement was recovered in analyses by Sallam *et al.* [66], using an independent phylogenetic data matrix.

A second postcranial skeleton of a sauropod was discovered in Zhejiang Province in 2007, this time from the centre of the province, in Dongyang City [62]. Based on an articulated vertebral sequence spanning most of the dorsal column through to the second caudal vertebra, as well as the pelvis, Lü *et al.* [61] described it the following year as a new taxon, *Dongyangosaurus sinensis*. These authors stated that it came from the early Late Cretaceous Fangyan Formation, but subsequent studies have shown that this unit is actually the Jinhua Formation ([63,67] and references therein). Lü *et al.* [61] considered *Dongyangosaurus* to be a 'basal' titanosaur, given that it lacks procoelous caudal vertebrae. Whereas D'Emic [40] optimized *Dongyangosaurus* as a euhelopodid somphospondylan, other authors have regarded it as a titanosaur, although its position is unstable. Mannion *et al.* [41] incorporated it into a phylogenetic analysis, in which it was recovered as a saltasaurid with a close relationship to *Opisthocoelicaudia skarzynskii*, from the Maastrichtian of Mongolia [68]. Based on the absence of caudal vertebral procoely, Averianov & Sues [51] questioned this placement and argued for a non-lithostrotian titanosaur placement, although it should be noted that *Opisthocoelicaudia* also lacks procoelous caudal vertebrae. However, subsequent iterations of the Mannion *et al.* [41] matrix have supported a 'basal' titanosaurian placement [69–71].

Here, we provide a detailed re-description of *Jiangshanosaurus lixianensis*, as well as new anatomical information on *Dongyangosaurus sinensis*, based on first-hand study. We use these amended data to re-examine the phylogenetic position of both taxa, evaluating whether or not they represent derived titanosaurs with close affinities to latest Cretaceous taxa. Finally, we present a new synthesis of the evolution and biogeographic history of Laurasian somphospondylans.

## 1.1. Institutional abbreviations

DYM, Dongyang Museum, Dongyang, Zhejiang, China; HBV, Shijiazhuang University Museum, Shijiazhuang, Hebei, China; MACN, Museo Argentino de Ciencias Naturales 'Bernardino Rivadavia', Buenos Aires, Argentina; MHNH, Museum d'histoire naturelle du Havre, France; NHMUK, Natural History Museum, London, United Kingdom; PMU, Palaeontological Museum, University of Uppsala, Sweden; ZMNH, Zhejiang Museum of Natural History, Hangzhou, Zhejiang, China.

# 2. Systematic palaeontology

SAUROPODA Marsh, 1878

MACRONARIA Wilson & Sereno, 1998

TITANOSAURIFORMES Salgado, Coria and Calvo, 1997

SOMPHOSPONDYLI Wilson & Sereno, 1998

*JIANGSHANOSAURUS* Tang *et al.*, 2001

**Type species:** *Jiangshanosaurus lixianensis*

**Holotype:** ZMNH M1322—five middle–posterior dorsal vertebrae, two anterior caudal vertebrae, one middle caudal vertebra, left scapulocoracoid, partial pubes and ischia, and shaft of left femur.

**Locality and horizon:** Lixian, Jiangshan County, Zhejiang Province, China; lower section of the Jinhua Formation, early Late Cretaceous.

**Revised diagnosis:** *Jiangshanosaurus lixianensis* can be diagnosed by two autapomorphies (marked with an asterisk), as well as five local autapomorphies: (i) spinodiapophyseal laminae absent in posterior dorsal vertebrae; (ii) centroprezygapophyseal fossa (CPRF) in anteriormost caudal vertebrae; (iii) ventral ends of spinoprezygapophyseal laminae situated medial to (rather than contacting) prezygapophyses in anterior–middle caudal vertebral transition*; (iv) dorsal margins of scapula and coracoid almost level, with no V-shaped gap; (v) coracoid glenoid does not curl upwards to expose the glenoid surface in lateral view; (vi) ridge for attachment of M. flexor tibialis internus III on ischium associated with groove; (vii) distal end of ischium terminates in a small hook-like dorsolateral process.*

**Table 1.** Measurements of vertebrae of *Jiangshanosaurus lixianensis* (ZMNH M1322). Cd A, B and C refer to the proximal anterior, distal anterior and proximal middle caudal vertebrae, respectively. Measurements in millimetres.

| dimension | Dv 10 | Dv 11 | CdA | CdB | CdC |
|---|---|---|---|---|---|
| centrum length (including condyle) | 291 | — | — | — | — |
| centrum length (excluding condyle) | 181 | 189 | 110 | 128 | 126 |
| anterior centrum width | — | — | 278 | 100 | 114 |
| anterior centrum height | — | — | 257 | 125 | 121 |
| posterior centrum width | 174 | — | 255 | 102 | 107 |
| posterior centrum height | 281 | 302 | 250 | 126 | 122 |
| neural arch height | ∼152 | — | — | 47 | 29 |
| neural spine height | 240 | — | — | 156 | — |
| neural spine mediolateral width (at base) | — | — | — | 34 | — |
| neural spine anteroposterior length (at base) | — | — | — | 62 | — |
| neural spine maximum mediolateral width | — | — | — | 40 | — |

# 3. Description and comparisons

## 3.1. Dorsal vertebrae

Five middle–posterior dorsal vertebrae (Dv) are preserved (see table 1 for measurements). Only one of the two more anteriorly positioned dorsal vertebrae, represented by a centrum, was figured by Tang *et al.* [60, pl. 1, fig. 8; pl. 2, figs 5, 7], who interpreted this as Dv 7. The three most posterior dorsal vertebrae were articulated at the time of discovery and were interpreted as Dv 9–11, but the anteriormost of these vertebrae was subsequently separated [60, pl. 1, figs 9–11]. Dv 9 preserves the centrum and lower neural arch. Dv 10 and 11 are relatively complete vertebrae (figure 1); however, preservation is poor in some places, especially along the neural spines and the posterior surface of Dv 11. Both vertebrae have also undergone a small amount of transverse compression, with the diapophyses crushed. Furthermore, the vertebrae are not fully prepared, meaning that their left surfaces cannot be observed. The approximate position of the remaining dorsal vertebra is unclear, but for simplicity we refer to it as Dv 8. Dv 7–9 are now incorporated into the mounted skeleton, and are fully restored, thereby greatly limiting their accessibility and anatomical utility. As such, most of our description is based on the two remaining articulated dorsal vertebrae that were originally identified as Dv 10 and 11.

All centra have a prominent anterior convexity that forms a sharp rim, separating the condyle from the remainder of the centrum (figure 1*a*). Each centrum is dorsoventrally taller than its transverse width (table 1). There are no ridges or fossae on the ventral surfaces, which are convex transversely. The lateral surfaces of the dorsal centra are excavated by a pneumatic opening that is set within a shallow fossa, as is the case in most somphospondylans [2]. These openings are biased towards the anterodorsal corner of the centrum. Although partially filled with matrix, these lateral pneumatic openings do not seem to ramify particularly deeply. There is evidence for a subhorizontal ridge inside at least some of these openings. Parapophyses are clearly absent from all five preserved centra, supporting the interpretation that these are not anterior dorsal vertebrae.

The anterior neural canal opening is set within a CPRF (figure 1*b*), as is the case in most eusauropods, with the exception of some titanosaurs, e.g. *Alamosaurus* and *Saltasaurus* [69,72]. Centroprezygapophyseal laminae (CPRLs) are non-bifid. Prezygapophyseal articular surfaces are largely flat and are tilted at least 30° to the horizontal. A steeply tilted zygapophyseal table is characteristic of the posterior dorsal vertebrae of titanosaurs, including putative forms such as *Baotianmansaurus* and *Ruyangosaurus* [69,72]. There is no hypantrum, which suggests that a hyposphene was probably absent, at least in posterior dorsal vertebrae, as is the case in most somphospondylans [1].

Although the parapophysis is not preserved on any vertebra, its position can be estimated based on the orientation of several parapophyseal laminae: it was probably situated at a similar height as the prezygapophyses in Dv 10 and 11, and possibly ventral to the diapophysis in Dv 11. The anterior centroparapophyseal lamina (ACPL) extends steeply anteroventrally to merge with the CPRL

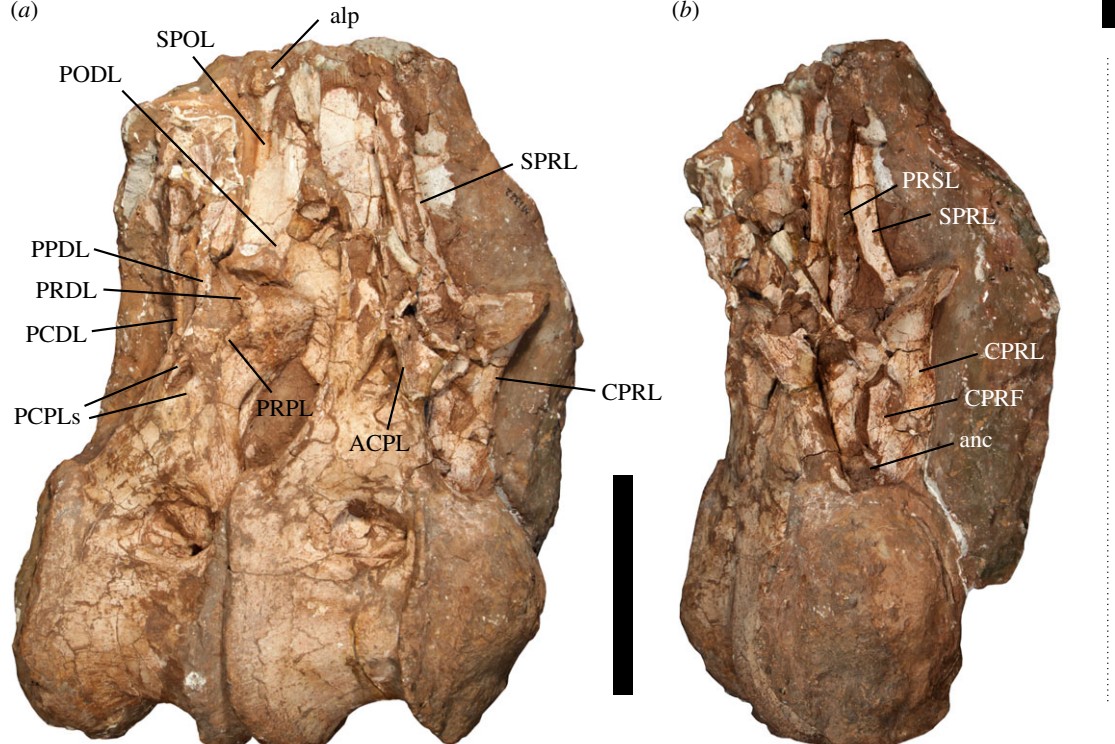

**Figure 1.** Posterior dorsal vertebrae (Dv 10–11) of *Jiangshanosaurus lixianensis* (ZMNH M1322) in: (*a*) right lateral and (*b*) anterior views. Abbreviations: ACPL, anterior centroparapophyseal lamina; alp, aliform process; anc, anterior neural canal opening; CPRF, centroprezygapophyseal fossa; CPRL, centroprezygapophyseal lamina; PCDL, posterior centrodiapophyseal lamina; PCPLs, posterior centroparapophyseal laminae; PODL, postzygodiapophyseal lamina; PPDL, paradiapophyseal lamina; PRDL, prezygodiapophyseal lamina; PRPL, prezygoparapophyseal lamina; PRSL, prespinal lamina; SPOL, spinopostzygapophyseal lamina; SPRL, spinoprezygapophyseal lamina. Scale bar equals 200 mm.

ventrally, although no prominent fossa is formed between these two laminae. Two posterior centroparapophyseal laminae (PCPLs) are present (figure 1*a*). The upper PCPL is a prominent lamina that is oriented steeply anterodorsally and presumably merges with the ACPL dorsally. By contrast, the lower PCPL is a much less well-developed ridge that is not as steeply oriented, and merges with the lower half of the ACPL. There is a shallow fossa on the lateral surface of the arch, anteroventral to the upper PCPL, as well as a more prominent fossa in between the two PCPLs, with the ACPL forming the anterodorsal margin of this excavation.

Both the posterior centrodiapophyseal lamina (PCDL) and paradiapophyseal lamina (PPDL) are subvertical (figure 1*a*), and demarcate a prominent, dorsoventrally elongate fossa. The PCDL does not notably widen or bifurcate at its ventral end. A poorly preserved prezygoparapophyseal lamina (PRPL) and prezygodiapophyseal lamina (PRDL) are discernible on Dv 11 (figure 1*a*). The diapophysis clearly projects strongly dorsolaterally, and a postzygodiapophyseal lamina (PODL) is still present, at least on Dv 10. A fossa is present between the PRPL, PPDL and the prominent spinoprezygapophyseal lamina (SPRL).

The neural spine projects mainly dorsally, with a slight posterior deflection. Although slightly incomplete, the neural spine is dorsoventrally short. In lateral view, the anteroposterior length of the neural spine appears to be consistent along its vertical extent. Although we cannot determine if there was any subtle bifurcation of the neural spine, there is clearly no deep division into separate metapophyses. A distinct, rugose prespinal lamina (PRSL) extends along the midline of the anterior surface of the neural spine (figure 1*b*), as is the case in most somphospondylans [41]. SPRLs are restricted to the anterolateral margins of the neural spine. In this regard, *Jiangshanosaurus* differs from many titanosaurs (e.g. *Alamosaurus* and *Opisthocoelicaudia*), in which the SPRLs are short and merge into the PRSL near to the base of the neural spine [69,72]. In contrast to the posterior dorsal vertebrae of nearly all other eusauropods [26], there is no evidence for spinodiapophyseal laminae (figure 1*a*). As such, we regard their absence as an autapomorphic reversal in *Jiangshanosaurus*. Weakly developed aliform processes project laterally near the neural spine apex. Little can be ascertained of the

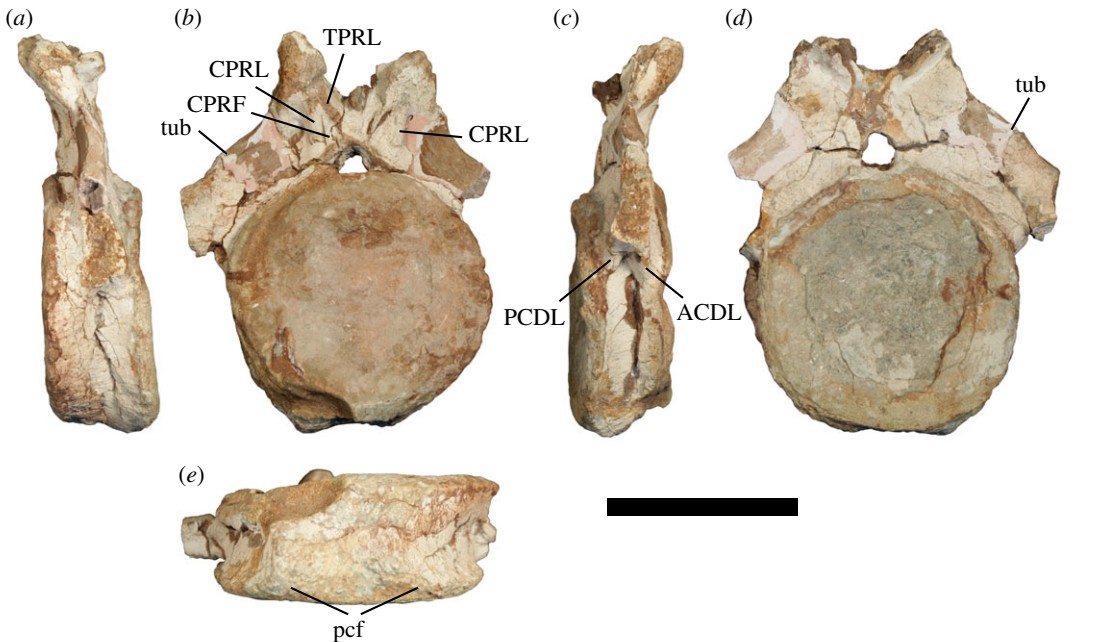

**Figure 2.** Proximal anterior caudal vertebra of *Jiangshanosaurus lixianensis* (ZMNH M1322) in: (*a*) left lateral, (*b*) anterior, (*c*) right lateral, (*d*) posterior and (*e*) ventral views. Abbreviations: ACDL, anterior centrodiapophyseal lamina; CPRF, centroprezygapophyseal fossa; CPRL, centroprezygapophyseal lamina; PCDL, posterior centrodiapophyseal lamina; TPRL, interprezygapophyseal lamina; tub, tubercle. Scale bar equals 200 mm.

morphology of the posterior surface of the neural arch and spine. It is also not possible to determine the nature of the internal tissue structure.

## 3.2. Caudal vertebrae

### 3.2.1. Proximal anterior caudal vertebra

The anteriormost preserved caudal vertebra (figured in [60, pl. II, figs 1, 2]) comprises the centrum, the base of the neural arch (including prezygapophyses), and the bases of the caudal ribs (figure 2; see table 1 for measurements). Tang *et al*. [60] interpreted this as the first caudal vertebra. Although we agree that it is from the proximal end of the tail, it is unlikely to be Cd1, based on the presence of chevron facets, which are usually absent from the first few caudal vertebrae [2]. The internal tissue structure of the vertebra is fine and spongey, as is also the case in the more distally preserved caudal vertebrae. This absence of camellae contrasts with the pneumatized anteriormost caudal vertebrae of several lithostrotian titanosaurs [26,41].

The centrum is anteroposteriorly short compared with its height and width, with an average elongation index (aEI) of 0.41. This value is lower than that of most other sauropods, which typically have values closer to 0.6 [2], and even reaching 0.92 in *Alamosaurus* [41]. However, the low aEI value of *Jiangshanosaurus* is similar to that of the East Asian somphospondylans *Baotianmansaurus* (0.45), *Opisthocoelicaudia* (0.46) and *Tangvayosaurus* (0.35) [41]. The centrum is slightly mediolaterally wider than dorsoventrally tall. Its ventral surface is flat to very mildly concave transversely, and lacks clearly defined ventrolateral ridges. Posterior chevron facets are present, but are poorly preserved. They are widely separated from one another, indicating that proximal chevrons were unlikely to have been dorsally bridged. The lateral surface of the centrum is anteroposteriorly concave and dorsoventrally convex, and lacks fossae, foramina and ridges.

The anterior articular surface of the centrum is concave, with a weakly developed, small central bump, and the posterior articular surface is consistently concave. Tang *et al*. [60] described the centrum as procoelous, which has led many subsequent authors to assume that there is a posterior convexity (e.g. [2,41]). However, the 'true' definition of procoely solely describes the concave nature of the anterior surface of the centrum [73], although the term has regularly been used to describe sauropod caudal centra that have a convex posterior articular surface (e.g. [1,23,74]). Regardless of the

usage of procoely intended by Tang *et al.* [60], we can unambiguously state that the preserved caudal centra of *Jiangshanosaurus* lack a posterior convexity. As such, the amphicoelous anterior caudal centra of *Jiangshanosaurus* contrast with those of nearly all titanosaurs [1], with the exception of *Savannasaurus* [69] and some putative titanosaurs (i.e. *Baotianmansaurus* and *Dongyangosaurus* [41]).

The caudal rib extends from the upper third of the centrum and onto the neural arch, and projects laterally. Although the ventral margin of the caudal rib is deflected dorsolaterally, only the base is preserved and so we cannot determine whether this orientation was maintained distally. The dorsal margin of the caudal rib faces dorsolaterally, lacking the 'fan'-shape that characterizes many diplodocoids [49,75]. Neither the anterior nor posterior surfaces of the caudal rib are excavated. A short, distinct anterior centrodiapophyseal lamina (ACDL) supports the caudal rib, and there is a PCDL too. These diapophyseal laminae are absent from the caudal vertebrae of most non-diplodocoids [26,49], although an ACDL is also present in a small number of brachiosaurids, as well as the basal somphospondylans *Phuwiangosaurus* and *Tastavinsaurus* [76]. It is not possible to determine whether a distinct PRDL is present because of preservation, but a PODL is definitely absent. Although incompletely preserved on both sides, a tubercle is situated on the dorsal surface of the caudal rib, close to the base of the prezygapophysis. A comparable tubercle is present on the anteriormost caudal vertebrae of a wide array of eusauropods [69], including numerous somphospondylan taxa [31].

The neural canal is wider than tall. Each prezygapophysis is supported ventrally by a thick, vertical CPRL. In anterior view, the interprezygapophyseal lamina (TPRL) is V-shaped, with the ventral tip of this 'V' meeting the roof of the anterior neural canal opening. The CPRL and TPRL form a subtriangular CPRF on the anterior surface of the neural arch. A CPRF is present in the anteriormost caudal vertebrae of several diplodocoids, but otherwise seems to be restricted to a small number of derived titanosaurs, e.g. *Saltasaurus* [71]. We therefore consider this feature to be a local autapomorphy of *Jiangshanosaurus*. The prezygapophyses project strongly dorsally, such that they do not extend beyond the anterior margin of the centrum, and their flat articular surfaces face dorsomedially. The posterior surface of the neural arch is poorly preserved and it is not possible to determine whether or not a hyposphene was present.

### 3.2.2. Distal anterior caudal vertebra

A vertebra from the distal end of the anterior caudal series (figured in [60, pl. II, figs 3, 4, 6, 8]) is mostly complete (see table 1 for measurements), although the left side of the centrum has been worn away, the left prezygapophysis is incomplete, and the right prezygapophysis has been distorted and displaced (figure 3a–e). The centrum is transversely compressed. It has a shallow midline ventral concavity, but no distinct ventrolateral ridges. The anterior articular surface of the centrum is irregular: overall, it is concave, with a small central bulge, but it forms a convexity along its dorsal third. By contrast, the posterior articular surface of the centrum is consistently concave. There are no openings or ridges on the lateral surface of the centrum, although there is a prominent bulge-like process on the arch-centrum junction. The caudal rib is incomplete (and only preserved on the right side), but is clearly reduced.

The neural arch is situated on the anterior two-thirds of the centrum. The neural canal is elliptical and taller than wide. Despite their distortion, the prezygapophyses clearly did not project far beyond the anterior margin of the centrum. The SPRLs extend down to the base of the prespinal fossa, such that they do not truly contact the prezygapophyses; this morphology is regarded as an autapomorphy of *Jiangshanosaurus*. Dorsally, the SPRLs fade out at about spine midheight and are restricted to the anterolateral margin of the neural spine. A ridge extends between the prezygapophysis and postzygapophysis at the base of the lateral surface of the neural spine, forming the floor of a shallow spinodiapophyseal fossa (SDF). A similar ridge is seen in the anterior–middle caudal transitional region of several brachiosaurids, as well as the somphospondylans *Andesaurus* and *Huabeisaurus* [31,77]. The postzygapophyses are large processes that border a prominent postspinal fossa at the base of the neural spine. Their articular surfaces are flat to very mildly convex, and face posteroventrally, as well as medially. The neural spine projects posterodorsally, extending to approximately midlength of the proceeding caudal vertebra, although its anterodorsal margin does not extend further posteriorly than the postzygapophyses. It is a transversely thin structure, although it thickens slightly dorsally, and its dorsal surface is convex both transversely and anteroposteriorly. There is no distinct prespinal ridge, and no clear evidence for a postspinal ridge.

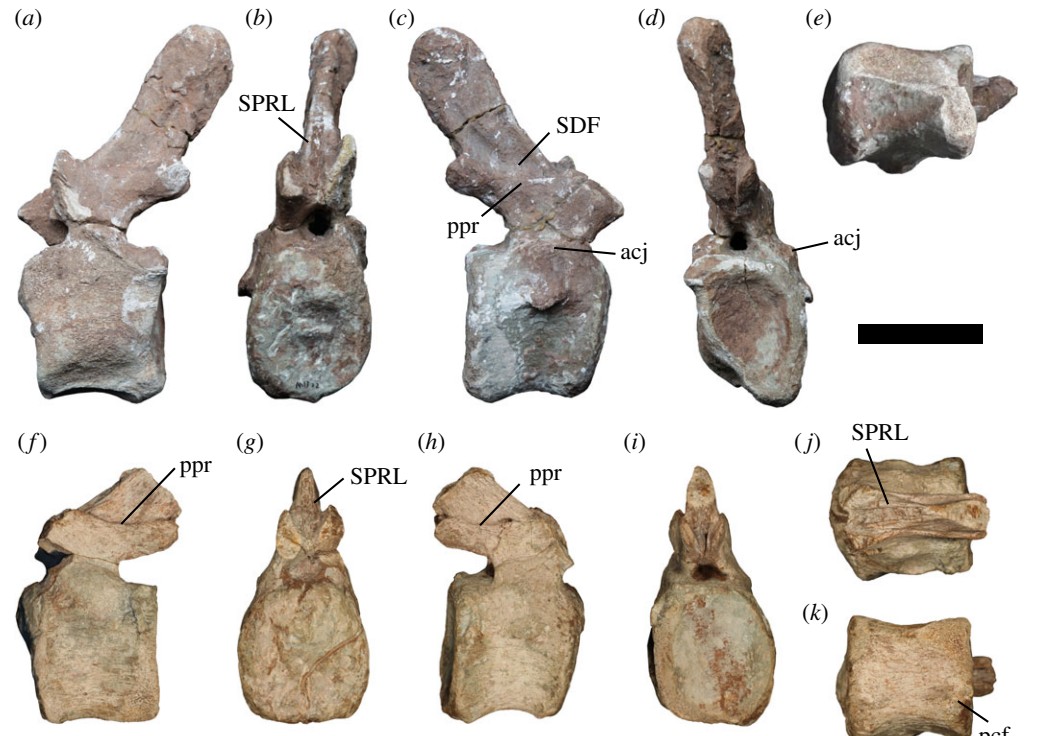

**Figure 3.** Distal anterior and proximal middle caudal vertebrae of *Jiangshanosaurus lixianensis* (ZMNH M1322). Anterior caudal vertebra in: (*a*) left lateral, (*b*) anterior, (*c*) right lateral, (*d*) posterior and (*e*) ventral views; middle caudal vertebra in: (*f*) left lateral, (*g*) anterior, (*h*) right lateral, (*i*) posterior, (*j*) dorsal and (*k*) ventral views Abbreviations: acj, arch-centrum junction; pcf, posterior chevron facet; ppr, prezygo-postzygapophyseal ridge; SDF, spinodiapophyseal fossa; SPRL, spinoprezygapophyseal lamina. Scale bar equals 100 mm.

### 3.2.3. Proximal middle caudal vertebra

A vertebra from the proximal region of the middle caudal vertebral series (figured in [60, pl. II, figs 9–11]) preserves the centrum and base of the arch, although the portion of prezygapophysis shown in the original images is no longer present (figure 3*f–k*). The centrum is slightly taller than wide (table 1). There are no ventrolateral ridges, and the ventral surface is only very mildly concave transversely. Posterior chevron facets are present and are well separated along the midline.

The anterior articular surface of the centrum is similar to that of the preceding caudal vertebra, being gently concave ventrally and convex along its dorsal third, whereas the posterior articular surface is more deeply concave. This condition, whereby the posterior articular surface is more deeply concave than the anterior surface, characterizes the anterior–middle caudal centra of several other mid–Late Cretaceous East Asian somphospondylans (*Gobititan*, *Huabeisaurus*, 'Huanghetitan' ruyangensis, *Phuwiangosaurus*, *Tambatitanis*, *Tangvayosaurus*), as well as the Australian titanosaur *Savannasaurus* [31,50,69]. The caudal rib is reduced such that it is now just a ridge a short distance below the dorsal margin of the centrum. There are no other ridges or fossae on the lateral surface of the centrum, although there is still a bulge on the arch–centrum junction.

The neural arch is anteriorly biased, and the prezygapophyses project anterodorsally. The autapomorphic SPRL morphology described in the preceding vertebra is again present, with the ventral ends of the SPRLs situated medial to the prezygapophyses. In addition, the ridge linking the prezygapophysis with the postzygapophysis is still present, forming a subtle shelf to a barely perceptible SDF. Postzygapophyses are still very prominent structures that form a moderately deep postspinal fossa at the base of the spine, and they extend beyond the posterior margin of the centrum. The postzygapophyseal articular surfaces are mildly convex. Based on its preserved base, the neural spine would have projected posterodorsally at approximately 45° to the horizontal.

### 3.3. Scapulocoracoid

The scapulocoracoid is here described with the long axis of the scapular blade oriented horizontally (figure 4*a,b*; see table 2 for measurements). With the exception of the distal portion of the blade, the

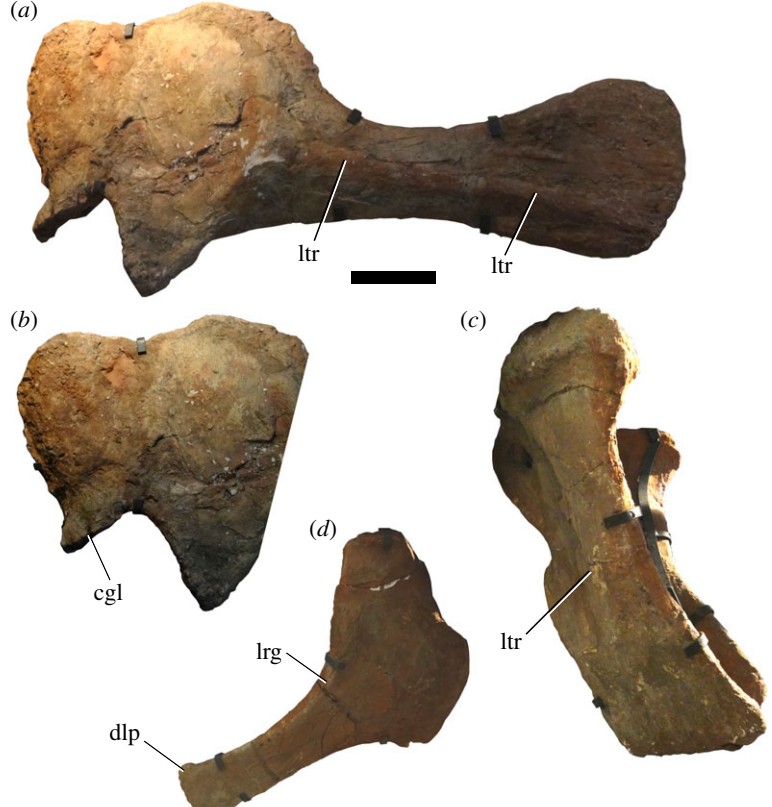

**Figure 4.** Appendicular elements of *Jiangshanosaurus lixianensis* (ZMNH M1322): (*a*) left scapulocoracoid in lateral view (coracoid is slightly oblique); (*b*) left coracoid and scapular acromion in lateral view; (*c*) pubes in right lateral view; and (*d*) right ischium in lateral view. Abbreviations: cgl, coracoid glenoid; dlp, dorsolateral process; lrg, lateral ridge for M. flexor tibialis internus III and associated groove; ltr, lateral ridge. Note that these specimens are all incorporated into the mounted skeleton. Scale bar equals 200 mm.

scapula is largely complete [60]. Although the coracoid was originally described as complete by Tang *et al.* [60], its anterior margin is poorly preserved and it is likely that some material is missing, especially at the anteroventral corner. This would also explain why it appears to be unusually short anteroposteriorly. Much of the medial surface of the scapulocoracoid is also coated with plaster. It is not possible to observe the internal tissue structure.

The scapula-coracoid articular surface is approximately 90° to the long axis of the blade. The dorsal margins of the scapula and coracoid are roughly level with one another, with only a shallow concavity, rather than a V-shaped notch, between them. This morphology is otherwise known only in titanosaurs [75], and is therefore regarded as a local autapomorphy of *Jiangshanosaurus*. As in all somphospondylans [74], the scapular glenoid is bevelled medially. The scapula also has a greater contribution to the glenoid than the coracoid. The region immediately posterior to the scapular glenoid forms a distinct ridge where the lateral surface meets the posteromedial surface of the acromion. This lateral ridge is not continuous with the ventral margin of the scapular blade. Instead, the ventral blade margin extends medial to the lateral ridge and fades out into the posteromedially facing flattened area that lies medial to the lateral ridge. Dorsally, the lateral ridge fades out into the posterolateral surface of the acromion, at approximately the level of the acromial ridge.

Anterior to the acromial ridge, the lateral surface of the acromion is fairly flat and featureless. The anteroposteriorly thick, low acromial ridge is subvertical (with a slight anterior deflection) relative to the long axis of the scapular blade, and there is no excavation of the lateral surface posterior to this ridge. The posterior margin of the dorsal third of the acromion is straight and slopes to face posterodorsally. There is no evidence for ventral tubercles on either the acromion or proximal part of the blade. The lateral surface of the scapular blade is dorsoventrally convex, forming a low, rounded ridge that extends posteroventrally along much of the length of the scapula. By contrast, the medial surface is fairly flat and there is no clear evidence for tubercles or ridges on this surface of the blade, but there is too much plaster to be entirely certain. The base of the scapular blade therefore has a D-shaped cross section. This morphology characterizes the scapulae of most

**Table 2.** Measurements of appendicular elements of *Jiangshanosaurus lixianensis* (ZMNH M1322). Measurements in millimetres.

| element and dimension | measurement |
| --- | --- |
| *left scapulocoracoid* | |
| anteroposterior length of scapula | 1377[a] |
| dorsoventral height of acromion | 735 |
| anteroposterior length of acromion | 460 |
| minimum dorsoventral height of scapular blade | 225 |
| maximum diameter of scapular glenoid | 282 |
| anteroposterior length of coracoid | 392[a] |
| dorsoventral height of coracoid | 569 |
| maximum diameter of coracoid glenoid | 205 |
| dorsoventral height of scapulocoracoid articular surface | 440 |
| *right pubis* | |
| maximum anteroposterior length of distal end | 314 |
| maximum mediolateral width of distal end | 78 |
| *right ischium* | |
| maximum proximodistal length | 796[a] |
| iliac peduncle mediolateral width | 75 |
| iliac peduncle anteroposterior length | 196 |
| minimum dorsoventral height of blade | 131 |
| distal end maximum dorsoventral height | 165 |
| distal end maximum mediolateral width | 38 |
| *left femur* | |
| shaft minimum circumference | 650 |
| mediolateral width of shaft | 265 |
| anteroposterior length of shaft | 118 |

[a]A measurement based on an incomplete element.

eusauropods, whereas this cross section is rectangular in many somphospondylans, including *Alamosaurus* [26]. The scapular blade of *Jiangshanosaurus* clearly expands dorsoventrally at its incomplete distal end.

The coracoid has a rounded anterodorsal corner in lateral view, but this might not be genuine because it is potentially incomplete. It is not possible to detect the position of the coracoid foramen, presumably because it has been filled with plaster. The coracoid glenoid does not expand laterally or curl upwards to expose the glenoid surface in lateral view. In this regard, the coracoid of *Jiangshanosaurus* is comparable to those of many non-neosauropods, as well as several derived titanosaurs [69], and is therefore regarded as a local autapomorphy. There is a concave notch-like area on the ventral margin of the coracoid, immediately anterior to the glenoid. The lateral surface lacks any distinct tubercles, but preservation is poor in places, meaning that we cannot be certain of their absence.

## 3.4. Pubis

Both pubes are preserved and incorporated into the mounted skeleton (figure 4*c*; see table 2 for measurements). Each element is incomplete proximally, and a large amount of material is missing from their posterior margins [60, pl. I, figs 2–5]. There is a low rounded ridge on the lateral surface of the middle third of both pubes. This ridge extends posteroventrally from a point close to the anterior margin. Unlike several titanosaurs [10,69,78], there is no groove anterior to this ridge. The distal ends of the pubes are slightly expanded anteroposteriorly relative to the main shaft, but there is no pubic 'boot'. There is also no notable transverse expansion of the distal end, resulting in a laminar blade that is comparable to those in most somphospondylans [3,69].

## 3.5. Ischium

Both ischia are also preserved and incorporated into the mounted skeleton (figure 4d; see table 2 for measurements). They are incomplete proximally and missing some of their margins [60, pl. 1, figs 6, 7]. Based on its preserved lower portion, the iliac peduncle is approximately twice as long anteroposteriorly as it is wide, but there is unlikely to have been a large ischial contribution to the acetabulum. There is a sharp ridge for attachment of M. flexor tibialis internus III on the dorsolateral margin of the proximal end of the shaft, defining the lateral margin of a broad and deep longitudinal groove. The ridge shows some signs of damage, which means that it might have been larger and more bulbous, and the associated groove less prominent. Nevertheless, the presence of a groove contrasts with the ischia of nearly all titanosauriforms [40,69], and is herein regarded as an autapomorphic reversal characterizing *Jiangshanosaurus*. The ridge projects mainly laterally, and would not have extended above the level of the main shaft, in contrast to the ischia of the somphospondylans *Huabeisaurus* [31] and *Wintonotitan* [79].

Poor preservation means that it is not possible to determine the nature of the upper symphysis of the paired ischia. The distal shafts of the paired ischia would have been closer to the coplanar condition when articulated, as is the case in most macronarians and rebbachisaurids [74,75]. These distal ends show little in the way of expansion relative to the rest of the shaft, with the exception of an autapomorphic small hook-like dorsolateral process. They have rugose terminal surfaces that are at least five times as wide transversely as dorsoventrally.

## 3.6. Femur

Only a portion of the shaft of the left femur is preserved, which is incorporated into the mounted skeleton. It has a transversely elongate, elliptical cross section, with its mediolateral width more than double that of its anteroposterior diameter (table 2). The anterior surface lacks the midline ridge (linea intermuscularis cranialis) that characterizes some derived titanosaurs [40,78]. A low, rounded fourth trochanter is present on the medial margin of the posterior surface, and is not visible in anterior view. However, this region is heavily plastered, and the fourth trochanter might be entirely reconstructed, as implied by Tang *et al*. [60, p. 279]: 'The position of the 4th trochanter emerged on the shaft cannot be judged'.

# 4. Additional anatomical information on *Dongyangosaurus sinensis*

Brief and limited access was granted to P.D.M. to study the type specimen of *Dongyangosaurus sinensis* (DYM 04888). Despite these restrictions, this still enabled an improved and revised understanding of the anatomy of *Dongyangosaurus* compared to that provided by Lü *et al*. [61]. Here, we present a brief update to Lü *et al*. [61].

Dorsal vertebrae are camellate, and comparable to most somphospondylans [74]. The ventral surfaces of the dorsal centra lack ridges or excavations, differing from the morphology of *Opisthocoelicaudia* [68] and *Diamantinasaurus* [80]. Each CPRL consists of two parallel laminae, at least in posterior dorsal vertebrae. Outside of Diplodocoidea, this morphology is uncommon [23], although a bifid CPRL characterizes the middle–posterior dorsal vertebrae of *Huabeisaurus* [31] and *Saltasaurus* [71]. PCPLs also comprise two parallel laminae. A PODL is present throughout the dorsal series, contrasting with some derived titanosaurs (e.g. *Alamosaurus*, *Opisthocoelicaudia*), in which the PODL disappears in more posterior dorsal vertebrae [1]. Dorsal neural spines project posterodorsally and all are bifid, although the depth of bifurcation decreases along the sequence. Bifid dorsal neural spines are characteristic of a wide array of eusauropods, including mamenchisaurids, turiasaurs, many diplodocoids, *Camarasaurus*, *Daxiatitan* and *Opisthocoelicaudia* [26,52,75]. SPRLs form the anterolateral margins of the neural spine metapophyses. A midline lamina extends dorsoventrally along the anterior surface of each metapophysis. An SPDL is present throughout the dorsal series. There is an anterior and posterior SPDL in posterior dorsal vertebrae, and it appears that the former is a 'captured' SPRL. Paired SPDLs also characterize several titanosaurs, including *Baotianmansaurus*, *Epachthosaurus* and *Saltasaurus* [69]. The posterior SPDL is not bifurcated. SPOLs are bifid, at least in posterior dorsal vertebrae. No additional anatomical information could be gleaned from the sacrum.

Both the anterior and posterior surfaces of the centra of caudal vertebrae 1 and 2 are gently concave. Neither centrum has a lateral or ventral excavation. The first caudal rib appears to have been crushed, such that the dorsal surface is visible in lateral view. As such, the 'anchor'-like morphology in Lü

*et al.* [61, pl. 2, fig. B] is the result of an anterolaterally curving rib, combined with a posterior expansion. The distal tip of the caudal rib appears to articulate with the ilium, although it is possible that this is the product of deformation. The posterior expansion of the rib is similar to the condition identified in the early-branching titanosaurs *Andesaurus* and *Epachthosaurus* [69,81].

The anterior two-thirds of the preacetabular process have been anteriorly displaced, giving the impression that the ilium is unusually anteroposteriorly long. Nevertheless, the ilium is an anteroposteriorly elongate, dorsoventrally low element. The preacetabulum is strongly flared laterally, but lacks a horizontal 'platform'. There is also no ventral 'kink' on the preacetabulum. The pubic peduncle has been displaced, and thus its orientation cannot be determined. There is no subtriangular fossa on its lateral surface, and the ischiadic peduncle lacks a lateral protuberance.

Only the left pubis is visible. The obturator foramen is fully ringed by bone and is elliptical, with its long axis approximately parallel to that of the pubic shaft. There is no ambiens process. The pubic shaft is strongly twisted relative to the proximal plate, although this has probably been accentuated by crushing. Regardless, it means that the lateral surface largely faces posteriorly, such that the view presented in Lü *et al.* [61, pl. 2, fig. A] is essentially of the anterolateral margin of the shaft. As such, the shaft is not as anteroposteriorly narrow as it appears, and has a more 'standard' morphology. The original figure also gives the impression that the left pubis has a distal anterior boot; however, this is likely to be either a broken, distorted piece of this element, or is part of the right pubis. There is no ridge on the lateral surface of the shaft, contrasting with the condition in several titanosaurs, including *Opisthocoelicaudia* and *Saltasaurus* [10].

As is the case with the pubis, the view presented in Lü *et al.* [61, pl. 2, fig. A] does not actually show the ischium in lateral view; instead, it shows the dorsolateral margin, meaning that the distal shaft is not as narrow as it appears. The anterodorsal corner of the ischiadic plate seems to form an upturned area, with a concave acetabulum in lateral view, as is the case in most titanosaurs [40]. Although it is probably accentuated by crushing, there is a very prominent ridge for the attachment of M. flexor tibialis internus III, with no associated groove. The ischia are preserved in articulation, demonstrating that their distal ends are almost certainly coplanar; however, they have undergone crushing and deformation, with the distal extremities seemingly smeared out as flanges. As is also the case in titanosaurs [26,75], as well as *Huabeisaurus* [31], there is no emargination distal to the pubic articulation, on the ventral margin.

# 5. Phylogenetic analysis and results

We revised the existing scores of *Jiangshanosaurus* and *Dongyangosaurus* in the most recent version of the Mannion *et al.* [41] data matrix, which comprises 117 OTUs scored for 542 characters [71]. These existing scores were based solely on the original publications of Tang *et al.* [60] and Lü *et al.* [61]. Based on our first-hand observations of these two taxa, we revised 67 and 42 characters for *Jiangshanosaurus* and *Dongyangosaurus*, respectively. We also augmented our scores for the late Early Cretaceous Chinese somphospondylan *Ruyangosaurus giganteus* following the additional material described from the type locality by Lü *et al.* [82]. A small number of existing character scores were also revised for other somphospondylan taxa, including new information on the cranial anatomy of the 'basal' lithostrotian *Malawisaurus dixeyi* [83]. These changes are all summarized in appendix A. We also added seven somphospondylan taxa to our matrix: (i) the Early Cretaceous French sauropod *Normanniasaurus genceyi* was scored based on Le Loeuff *et al.* [46] and personal observations of the type material (MHNH-2013.2.1) by P.D.M. in 2019; (ii) *Europatitan eastwoodi*, from the Early Cretaceous of Spain, was scored based on Torcida Fernández-Baldor *et al.* [84]; (iii) the Early Cretaceous Chinese sauropod *Yongjinglong datangi* was scored based on Li *et al.* [54]; (iv) the Late Cretaceous Chinese taxon *Huabeisaurus allocotus* was scored based on D'Emic *et al.* [31] and personal observations of the type material HBV-20001 by P.D.M. and P.U. in 2012; (v) the Late Cretaceous Argentinean sauropod *Antarctosaurus wichmannianus* was scored based on von Huene [13] and personal observations of the type specimen MACN 6904 (cranial and mandibular material only) by P.D.M. (2013, 2018) and P.U. (2013); (vi) *Jainosaurus septentrionalis*, from the latest Cretaceous of India, was scored following Wilson *et al.* [85,86]; and (vii) *Vahiny depereti*, from the latest Cretaceous of Madagascar, was scored based on Curry Rogers & Wilson [87]. Six characters were also added (see appendix A): five new characters based on a review of the literature, and one modified from Santucci & Arruda-Campos [88]. The revised data matrix comprises 124 OTUs scored for 548 characters.

We followed the analytical protocol implemented in Mannion *et al.* [71]. Characters 11, 14, 15, 27, 40, 51, 104, 122, 147, 148, 195, 205, 259, 297, 426, 435, 472 and 510 were treated as ordered multistate characters, and several unstable and fragmentary taxa were excluded from the analyses *a priori*

(*Astrophocaudia*, *Australodocus*, *Brontomerus*, *Fukuititan*, *Fusuisaurus*, *Liubangosaurus*, *Malarguesaurus*, *Mongolosaurus*). Using equal weighting of characters, this pruned data matrix was analysed using the 'Stabilize Consensus' option in the 'New Technology Search' in TNT v. 1.5 [89,90]. Searches employed sectorial searches, drift and tree fusing, with the consensus stabilized five times, prior to using the resultant trees as the starting topologies for a 'Traditional Search', using Tree Bisection-Reconstruction. We then re-ran the analysis, using the same pruned matrix and protocol, but also applying extended implied weighting in TNT [91,92]. This approach downweights characters with widespread homoplasy during the tree search, with a concavity (*k*) value used to define the strength of downweighting. The lower the *k*-value, the more strongly a highly homoplastic character is downweighted [91]. Previous analyses of this dataset have used a *k*-value of 3, which is the default value in TNT. However, this is quite a severe application of downweighting, and simulations indicate that a higher value might be more appropriate [92]; see also [93]. As such, here we ran two sets of extended implied weighting analysis, using a *k*-value of 3 and 9. The revised data matrix is provided as both a nexus and TNT file (electronic supplementary material).

Our equal weights parsimony (EWP) analysis resulted in 792 MPTs of length 2654 steps. The strict consensus is well-resolved (figure 5*a*), and the overall topology is similar to that in Mannion *et al.* [71], albeit with greater resolution within Titanosauria. Bremer supports have values of 1 or 2 for most nodes. Analysis using extended implied weights, with a *k*-value of 3 (EIW3), produced 2376 MPTs of length 242.6 steps. Although the rest of the topology (figure 5*b*) is similar to that presented in Mannion *et al.* [71], there are a number of differences within Somphospondyli, as well as a large polytomy at the base of this clade. The Pruned Trees and Agreement subtree options in TNT show that this polytomy can be resolved by excluding two (*Padillasaurus* and *Sauroposeidon*) out of the 12 OTUs (note that *Sauroposeidon* is probably the senior synonym of *Paluxysaurus* [94], which is retained as a separate OTU in this resolved polytomy). When a *k*-value of 9 was used (EIW9), our analysis resulted in 5940 MPTs of length 137.9 steps. Overall, this topology (figure 5*c*) is closer to that of EIW3 than EWP, but there are a number of differences. As with the aforementioned analysis, a large polytomy close to the base of Somphospondyli is resolved through the pruning of *Padillasaurus* and *Sauroposeidon*.

# 6. Discussion

## 6.1. Are *Jiangshanosaurus* and *Dongyangosaurus* derived titanosaurs?

The results from our analyses of the revised data matrix consistently place *Jiangshanosaurus* and *Dongyangosaurus* in a more 'basal' position than previous iterations, without close affinities to the latest Cretaceous derived titanosaurs *Alamosaurus* and *Opisthocoelicaudia*. *Jiangshanosaurus* is placed in the endemic East Asian somphospondylan clade Euhelopodidae (i.e. outside Titanosauria) in the EWP analysis (figure 5*a*), but clusters with the Australian taxon *Diamantinasaurus*, as a lithostrotian titanosaur, in the EIW3 analysis (figure 5*b*). In our EIW9 topology, *Jiangshanosaurus* and *Dongyangosaurus* form a clade that is the sister taxon to most other titanosaurs, with *Huabeisaurus* the successive outgroup (figure 5*c*). *Dongyangosaurus* and *Huabeisaurus* form a clade in our EWP analysis, with this grouping being the sister taxon to Titanosauria (figure 5*a*). Both taxa are part of a paraphyletic array of 'basal' somphospondylans in our EIW3 analysis, although the stemwards shift of *Andesaurus* means that nearly all somphospondylans are recovered within Titanosauria in this topology (see also previous iterations of this matrix), as well as that of our EIW9 analysis (figure 5*b,c*). Below, our use of the clade name Titanosauria refers to the topology recovered in our EWP analysis.

As detailed above, very little of the anatomy of *Jiangshanosaurus* supports 'derived' titanosaurian affinities. Most features are plesiomorphic for Titanosauriformes or Somphospondyli, especially in the vertebrae. The one exception is that the dorsal margins of the scapula and coracoid are approximately level with one another in *Jiangshanosaurus* (figure 4*a,b*), a feature that is otherwise restricted to Titanosauria [75]. All of the features uniting *Jiangshanosaurus* with *Diamantinasaurus* are optimized either as reversals to the plesiomorphic state, and/or also characterize taxa outside of Titanosauria too.

*Dongyangosaurus* shares some features with titanosaurs, including the posterior expansion of the first caudal rib [69], and a strongly concave acetabular margin on the ischium [40]. Other anatomical features generally have a wider distribution among Somphospondyli. Its position as close to the titanosaur radiation is therefore in keeping with this character combination.

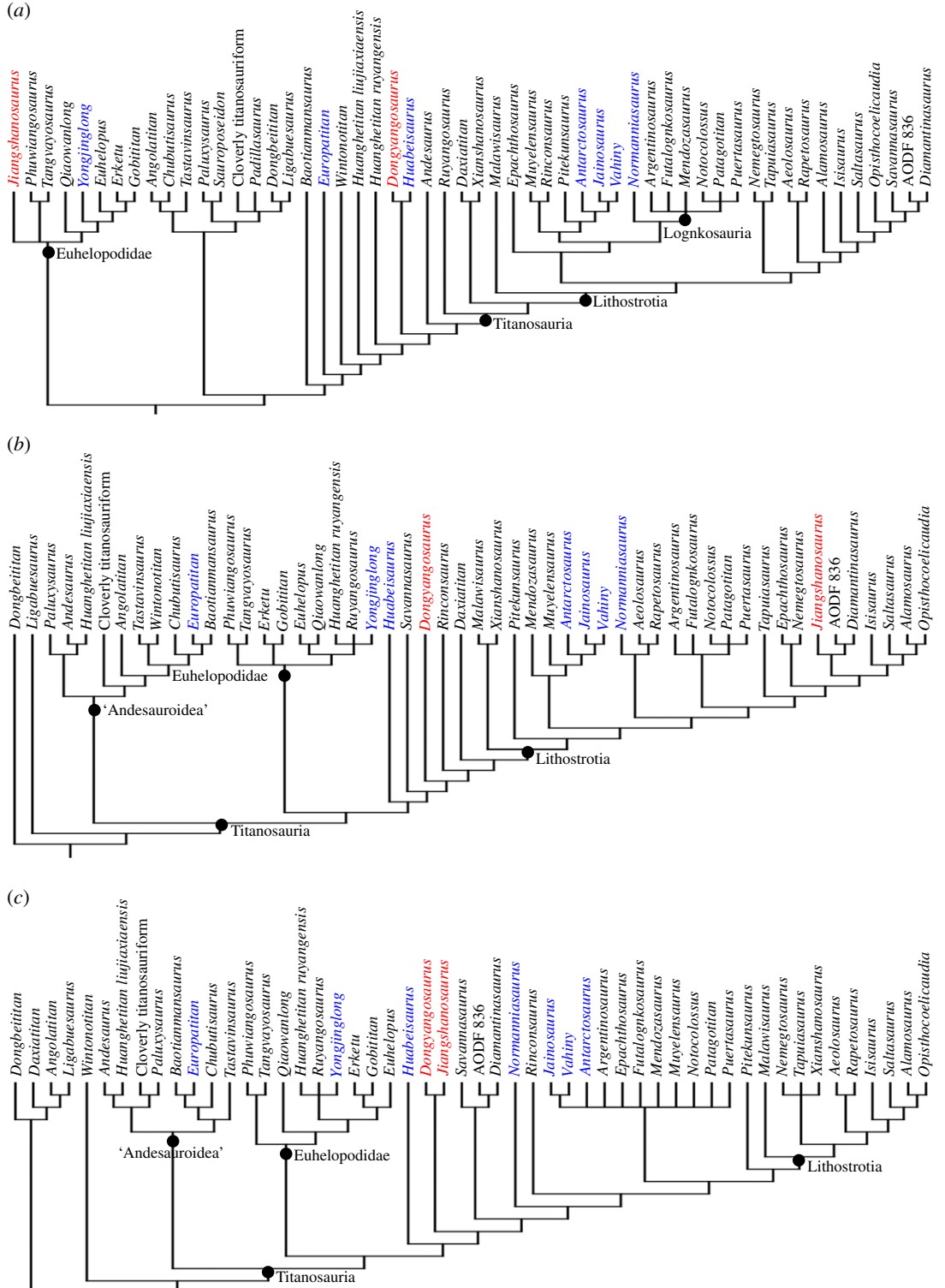

**Figure 5.** Strict consensus cladogram using (*a*) equal weights; (*b*) extended implied weights with a *k*-value of 3 and (*c*) extended implied weights with a k-value of 9. In all cases, only Somphospondyli is shown and each tree was produced following the *a priori* exclusion of seven unstable taxa (see text for details). In parts *b* and *c*, two further OTUs (*Padillasaurus*, *Sauroposeidon*) are pruned out (based on the agreement subtree) to resolve the polytomy near the base of the tree (the 'Andesauroidea' clade). *Jiangshanosaurus* and *Dongyangosaurus* are highlighted in red, and those taxa newly added to the data matrix are highlighted in blue.

## 6.2. The evolutionary and biogeographic history of Laurasian somphospondylans

There are no unambiguous occurrences of pre-Cretaceous somphospondylans, although the Tithonian Tanzanian titanosauriform *Australodocus bohetii* probably belongs to this clade [71]. Mocho *et al.* [95]

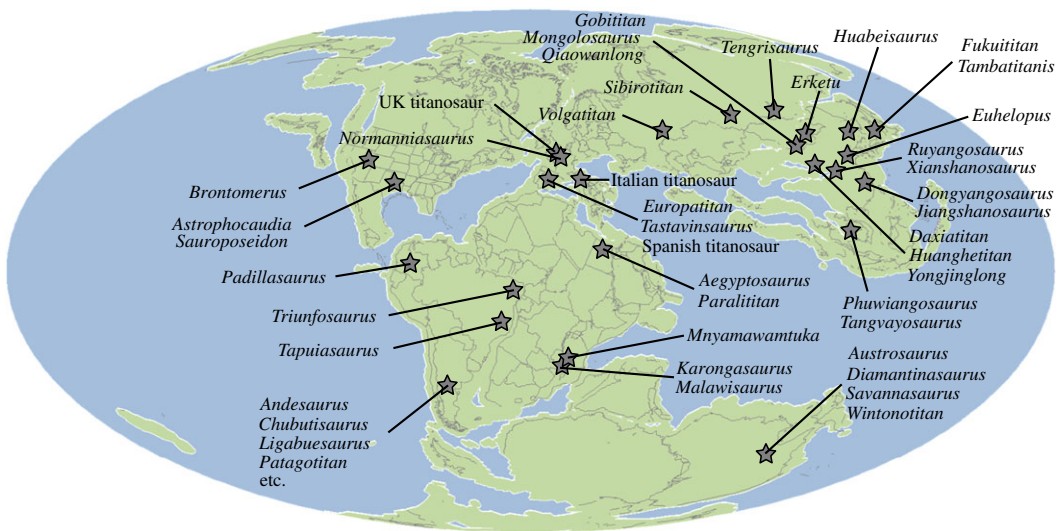

**Figure 6.** Palaeogeographic reconstruction showing the global distribution of somphospondylan titanosauriform sauropods in the Early Cretaceous–Cenomanian (reconstruction at 130 Ma). Note that this is not comprehensive. Reconstruction from Fossilworks (http://fossilworks.org/).

described the macronarian *Oceanotitan dantasi* from the late Kimmeridgian–early Tithonian of Portugal, and recovered it as the earliest diverging member of Somphospondyli in one of their phylogenetic analyses. Given that the somphospondylan sister clade Brachiosauridae is represented in the Late Jurassic (late Oxfordian–Tithonian) of western Europe, Tanzania and the USA, at least [76], Somphospondyli must also have diverged by the early Late Jurassic. As such, the possible presence of the clade in the Late Jurassic of Portugal and Tanzania would not be unexpected. Unequivocal somphospondylan occurrences are known from the earliest Cretaceous (figure 6), with *Triunfosaurus leonardii* from the Berriasian–early Hauterivian of Brazil [36], and *Euhelopus zdanskyi* from the Berriasian–Valanginian [96] of China [97]. Fragmentary remains from approximately contemporaneous Japanese deposits might also represent somphospondylans (e.g. [98]). No further somphospondylans are known from pre-Barremian deposits, but their absence might reflect the global scarcity of Berriasian–Hauterivian-aged terrestrial sedimentary rocks [99,100], rather than a genuine pattern.

By the Barremian–Aptian, somphospondylans had acquired a near-global distribution, including multiple taxa in the USA, across Europe and in East Asia (figure 6). All of the taxa known from the Barremian–Albian of the USA (*Astrophocaudia slaughteri*, *Brontomerus mcintoshi*, *Sauroposeidon proteles* [='*Paluxysaurus jonesi*']) are 'basal' somphospondylans [40,94], that lie outside of Titanosauria in our EWP analyses (figure 5a). Within Somphospondyli, the phylogenetic affinities of these North American taxa are poorly constrained. *Astrophocaudia* and *Brontomerus* are highly unstable [41,94,101], and different analyses result in *Sauroposeidon* clustering with a globally distributed array of taxa (e.g. figure 5). Of note is that the contemporaneous OTU comprising a somphospondylan from the Cloverly Formation ('Cloverly titanosauriform') is recovered as the sister taxon to *Sauroposeidon* in our EIW9 analysis (figure 5c), which would support the referral of this material to that taxon by D'Emic & Foreman [102]. Stratigraphically younger sauropods are unknown from North America until the Maastrichtian, with the appearance of *Alamosaurus sanjuanensis* ([65,103]; though see Ryan & Evans [104] for a possible Santonian sauropod occurrence from Canada), although whether this 'sauropod hiatus' reflects an extinction followed by 're-invasion', a sampling bias or some combination of both, remains uncertain [9,105].

The late Barremian–early Aptian Spanish sauropods, *Europatitan eastwoodi* and *Tastavinsaurus sanzi*, are recovered as 'basal' somphospondylans here (figure 5), as has been proposed in several other analyses [40,41,84,106]. In agreement with Torcida Fernández-Baldor *et al.* [84], our results suggest that they are not sister taxa, and possibly not closely related to one another (figure 5a). As is the case with contemporaneous North American somphospondylans, these Spanish genera cluster with taxa with a near-global distribution. Additional Barremian-aged remains come from the UK and include: (i) a cervical vertebra that potentially comes from a sauropod with close affinities to *Sauroposeidon* [41,107]; and (ii) caudal vertebrae that represent the earliest known occurrence of Titanosauria in western Europe [39,40,44] (figure 6).

A small number of additional occurrences provide further support for the presence of titanosaurs in the late Early Cretaceous to Cenomanian of western Europe [39,42]. These include a caudal vertebra from the late Aptian–early Albian of Italy, and postcrania from the Cenomanian of Spain (figure 6), both of which have been suggested to be allied with Gondwanan titanosaurs [42,47]. Le Loeuff *et al.* [46] erected *Normanniasaurus genceyi* as a 'basal' titanosaur from the Albian of France. Titanosaurian affinities for *Normanniasaurus* have subsequently been supported through phylogenetic analyses [30,42,108], although our study is the first to incorporate it based on first-hand observations. All three of our analyses place *Normanniasaurus* with Gondwanan titanosaurian taxa, clustering either with: (i) Lognkosauria (EWP) (figure 5a), in which it is the sister taxon to this clade of Late Cretaceous Argentinean titanosaurs [5,70]; (ii) Aeolosaurini (EIW3) (figure 5b), in which it forms a clade with *Aeolosaurus*+*Rapetosaurus krausei*, from the latest Cretaceous of South America [10] and Madagascar [25], respectively; or (iii) as the sister taxon to most other titanosaurs (EIW9; figure 5c). An aeolosaurine position is similar to that found by Gorscak *et al.* [30], and some independent phylogenetic analyses recover a clade containing Lognkosauria and Aeolosaurini (e.g. [5,109]). In both the EWP and EIW3 scenarios presented here, these two groups are nested within a large clade of Gondwanan taxa. This includes the latest Cretaceous Indo-Madagascan taxa *Jainosaurus septentrionalis* and *Vahiny depereti*, which are sister taxa, forming a clade with the approximately contemporaneous Argentinean taxon *Antarctosaurus wichmannianus* (figure 5a,b), as suggested by previous authors [70,85,87]. In our EIW9 topology (figure 5c), the clade that includes Lognkosauria and *Epachthosaurus sciuttoi* lies outside of Lithostrotia, whereas these taxa are lithostrotians in our other two analyses (and previous iterations of this matrix). A similar result, in which Lognkosauria and Lithostrotia are essentially sister clades, was recovered by Carballido *et al.* [5].

Three Early Cretaceous somphospondylan taxa have recently been named from across present-day Russia (figure 6), although all are based on relatively incomplete material. *Sibirotitan astrosacralis* is from the Barremian–Aptian of western Siberia, and appears to represent a non-titanosaurian somphospondylan [110]. Contemporaneous remains from south-central Russia were described as *Tengrisaurus starkovi* by Averianov & Skutschas [45], who recovered this taxon within Lithostrotia. It is not currently clear to which taxa *Sibirotitan* and *Tengrisaurus* are most closely related [43,45,110]. Stratigraphically pre-dating these occurrences, Averianov & Efimov [43] erected *Volgatitan simbirskiensis* from the late Hauterivian of western Russia. *Volgatitan* was placed close to the 'base' of the clade that includes Lognkosauria and Rinconsauria in that study, suggesting affinities with Gondwanan taxa [43]. Although *Normanniasaurus* also clusters with Lognkosauria in our EWP analysis, it appears to represent a lineage that is clearly distinct from *Volgatitan*. As such, it appears that titanosaurs were present across western Eurasia from the late Hauterivian–Barremian (approx. 130 Ma) onwards, and that these Eurasian lineages lie within a largely Gondwanan clade. Several authors have suggested that these Early Cretaceous–Cenomanian European somphospondylans can be explained by dispersal between north Africa and southern Europe (e.g. [38,42,47,84,95,111,112]), with some palaeogeographic support for a land connection, known as the Apulian Route, at least during the Berriasian–Barremian [113–115] or Hauterivian–Aptian [112]. Post-Cenomanian, there are no sauropod body fossils in Europe until the Santonian; this absence mirrors that of the North American 'hiatus' and most likely reflects the dearth of suitable terrestrial sedimentary rocks during this interval [39,105,116].

Although our EWP analysis suggests that neither *Dongyangosaurus* nor *Jiangshanosaurus* are titanosaurs, results from both of our EIW analyses mean that their titanosaurian affinities remain equivocal. Following our augmented scoring, *Ruyangosaurus giganteus* is a 'basal' titanosaur in the EWP analysis (figure 5a), and a euhelopodid (*sensu* [40]) in the EIW analyses (figure 5b,c). Li *et al.* [54] recovered *Yongjinglong datangi* as a titanosaur in three independent phylogenetic analyses, including an earlier iteration of the matrix used here [41]. By contrast, all of our analyses support a placement within Euhelopodidae (figure 5). This clade is diverse in all analyses, with a further six 'middle' Cretaceous East Asian taxa consistently included (*Erketu ellisoni*, *Euhelopus zdanskyi*, *Gobititan shenzhouensis*, *Phuwiangosaurus sirindhornae*, *Qiaowanlong kangxii* and *Tangvayosaurus hoffeti*). Most of these taxa have been recovered in Euhelopodidae in previous analyses too (e.g. [40,41]). Our new topologies also suggest that *Baotianmansaurus henanensis*, *Huanghetitan liujiaxiaensis* and '*Huanghetitan*' *ruyangensis* are 'basal' somphospondylans too (figure 5), and it seems likely that other contemporaneous East Asian taxa (i.e. *Borealosaurus wimani*, *Dongbeititan dongi*, *Fukuititan nipponensis*, *Fusuisaurus zhaoi*, *Liaoningotitan sinensis*, *Liubangosaurus hei*, *Mongolosaurus haplodon*, *Tambatitanis amicitiae*, *Yunmenglong ruyangensis*) also occupy a similar part of the tree [40,41,50,51,71,117]. Some of these might represent additional euhelopodids [40,50,117]. Based on our current knowledge, Euhelopodidae seems to have been endemic to East Asia, although there are remains from the Late

Jurassic (*Oceantotitan*; [95]) and Barremian (teeth; [118]) of western Europe, and Tithonian of Tanzania (*Australodocus*; [71]), that share some features with members of this clade.

Although it seems that most 'middle' Cretaceous Asian sauropods are 'basal' somphospondylans, one late Early Cretaceous Chinese sauropod is recovered within Titanosauria in all three analyses, and another in two analyses. *Daxiatitan binglingi* (Barremian–Aptian) and *Xianshanosaurus shijiagouensis* (Aptian–Albian [119]) are sister taxa, just outside of Lithostrotia, in our EWP analysis (figure 5*a*). *Daxiatitan* occupies the same position in our EIW3 analysis, whereas *Xianshanosaurus* is recovered as a 'basal' lithostrotian in both EIW analyses (figure 5*b,c*). By contrast, *Daxiatitan* is placed outside of Titanosauria in our EIW9 analysis (figure 5*c*). Combined with the caudal vertebra PMU 24709, from the Aptian–Albian of China [55], which most likely represents a titanosaur [41,49], these taxa suggest that titanosaurs had dispersed into East Asia by the Barremian–Aptian (approx. 129–113 Ma). Moreover, a number of fragmentary and isolated remains from Central Asia (Kazakhstan, Kyrgyzstan, Tajikistan and Uzbekistan) document the continued presence of probable titanosaurs in the early Late Cretaceous of Eurasia [51,120].

Previous iterations of the data matrix used herein (e.g. [41,71]) have consistently recovered *Jiangshanosaurus* as the sister taxon of *Alamosaurus*, with a close relationship also proposed in the original publication of Tang *et al.* [60]. *Alamosaurus* is known from the Maastrichtian of the USA [14,64], and is currently the only recognized titanosaur from North America [65,121]. Its biogeographic origin has long been the subject of debate [9,74,103,105,122], with alternative phylogenetic hypotheses supporting dispersal from either: (i) South America, based on a sister taxon relationship with latest Cretaceous taxa such as *Baurutitan britoi* or *Saltasaurus loricatus* (e.g. [1,38,40]); or (ii) East Asia, based on a sister taxon relationship with *Opisthocoelicaudia skarzynskii* (e.g. [74,109]), from the Maastrichtian of Mongolia [68]. Our previous result of a *Jiangshanosaurus+Alamosaurus* clade therefore led to the suggestion that the *Alamosaurus* lineage dispersed from East Asia at some point during the Late Cretaceous [41]. Revision of *Jiangshanosaurus* has removed this relationship, but does not resolve the biogeographic ancestry of the *Alamosaurus* lineage: our EWP analysis finds *Alamosaurus* nested with Gondwanan taxa (figure 5*a*), whereas our EIW topologies restore *Opisthocoelicaudia* as its sister taxon (figure 5*b,c*). Additional, well-preserved specimens of *Alamosaurus* might eventually shed light on its affinities [121,122]. Despite being one of the best-known titanosaurs, *Opisthocoelicaudia* has not been included in a phylogenetic analysis based on first-hand study, and many anatomical features cannot be adequately assessed from the photographs and illustrations in the original and sole publication [68]. The recent rediscovery of the type locality of the contemporaneous titanosaur *Nemegtosaurus mongoliensis* has added postcranial remains to a taxon previously known only from its skull [27,123]: these have yet to be fully described [124]. Whereas some authors have suggested that *Nemegtosaurus* might be synonymous with *Opisthocoelicaudia* (e.g. [124]), these new specimens, as well as additional remains from nearby coeval deposits (including *Quaesitosaurus orientalis*; [125]), suggest that they represent distinct taxa [126]. Although fragmentary, *Qingxiusaurus youjiangensis* [17], *Sonidosaurus saihangaobiensis* [127] and *Zhuchengtitan zangjiazhuangensis* [18] also point to a higher diversity of latest Cretaceous East Asian titanosaurs. Further study of these taxa will be critical in resolving the roles of East Asia versus South America in the appearance of *Alamosaurus* in the latest Cretaceous of the USA.

There is a rich record of titanosaurs in the late Campanian–Maastrichtian of Europe, primarily from France, Spain and Romania [115,128], although most taxa have only been recognized in the last two decades [129]. *Magyarosaurus dacus*, from the early Maastrichtian of Romania [12], was the first to be described, although it is taxonomically problematic, given that it is unclear which skeletal remains are referable [19]. The recognition of a second contemporaneous Romanian titanosaur, *Paludititan nalatzensis*, further complicates this situation [19]. *Ampelosaurus atacis* and *Atsinganosaurus velauciensis* have been described from the early Maastrichtian and late Campanian–early Maastrichtian, respectively, of France [15,20]. Two titanosaurs have also been named from Spain: the late Campanian taxon *Lirainosaurus astibiae* [16], and *Lohuecotitan pandafilandi*, from the late Campanian–early Maastrichtian [21]. These European taxa are generally underrepresented in most phylogenetic analyses, with none incorporated in either Carballido *et al.* [5] or the data matrix used here. By contrast, with the exception of *Magyarosaurus*, all of them were included in both Díez Díaz *et al.* [108] and Sallam *et al.* [66]; see also [130]. However, the former consists of a data matrix comprising 29 taxa scored for just 77 characters, and none of the European taxa was scored based on first-hand observations in the analysis of Sallam *et al.* [66]. Nevertheless, Díez Díaz *et al.* [108] erected the 'derived' titanosaur clade Lirainosaurinae to group *Lirainosaurus* and the two French taxa, which was nested among primarily Gondwanan taxa. They also recovered *Lohuecotitan* and *Paludititan* as sister taxa, but close to the 'base' of Lithostrotia. The

analyses of Sallam *et al.* [66] and Gorscak & O'Connor [130] separated *Atsinganosaurus* from the other European taxa, placing it within Lognkosauria. The remaining European taxa were recovered in a clade of predominantly Laurasian taxa (also containing *Nemegtosaurus* and *Opisthocoelicaudia*), although this also included the middle Campanian Egyptian titanosaur *Mansourasaurus shahinae*. *Alamosaurus* and several South American taxa either formed the sister taxon to this clade [66], or were nested within it [130]. Several authors have suggested that these latest Cretaceous titanosaurs dispersed into Eurasia from Africa (e.g. [39,66,131]), possibly via a re-emergent Apulian Route [114].

In summary, Somphospondyli must have evolved by the early Late Jurassic. Given that we have good evidence for the clade in the earliest Cretaceous of Brazil and China (and possibly in the Late Jurassic of Portugal and Tanzania), and elsewhere from the Barremian–Aptian onwards, this suggests that the group had a near-global distribution early in its known evolutionary history (figure 6). It is possible that much of this is currently unsampled. The earliest unambiguous titanosaurian remains are late Hauterivian in age, and the clade was present in Africa, Asia, Europe and South America by the Aptian (figure 6). As such, the evidence points to the diversification and widespread distribution of titanosaurs by at least approximately 130–120 Ma [38,43,69]. Given that the oldest known remains are Eurasian, rather than Gondwanan, the origin and early evolutionary history of Titanosauria remains uncertain. Many Laurasian titanosaurs appear to be closely related to Gondwanan taxa, but much of their biogeographic history is currently unclear. Further study and incorporation of additional Laurasian taxa into large-scale, global phylogenetic analyses, as well as the revision and/or augmentation of key taxa (e.g. *Nemegtosaurus*, *Opisthocoelicaudia*), is therefore critical to elucidating the evolutionary relationships and biogeographic history of titanosaurs.

# 7. Conclusion

A full re-description of the early Late Cretaceous titanosauriform sauropod *Jiangshanosaurus lixianensis*, previously regarded as a derived titanosaur, demonstrates that it almost entirely lacks titanosaurian features, and is most likely a 'basal' member of Somphospondyli. New anatomical information on the contemporaneous taxon *Dongyangosaurus sinensis* suggests that it also lies outside of Titanosauria, although it is probably closer to this radiation than *Jiangshanosaurus*. Although most other 'middle' Cretaceous sauropods from East Asia are also probably non-titanosaurian somphospondylans, at least two genera (*Daxiatitan* and *Xianshanosaurus*) appear to belong to the titanosaur radiation. Combined with these and other approximately contemporaneous European titanosaurs, the recovery of the late Early Cretaceous French sauropod *Normanniasaurus genceyi* as a 'derived' titanosaur, nested with Gondwanan taxa, provides further support for a widespread distribution of this clade by the Early Cretaceous.

Data accessibility. The datasets supporting this article have been uploaded as part of the electronic supplementary material.

Authors' contributions. P.D.M. conceived of, designed and coordinated the study, collected and interpreted the data, performed the analyses and drafted the manuscript and figures. P.U. participated in the design of the study, collected and interpreted data and critically revised the manuscript. X.J. provided access to materials. W.Z. provided access to materials, took photographs and critically revised the manuscript. All authors gave final approval for publication and agree to be held accountable for the work performed therein.

Competing interests. We declare we have no competing interests.

Funding. P.D.M.'s research was funded by a Leverhulme Trust Early Career Fellowship (ECF-2014-662) and a Royal Society University Research Fellowship (UF160216). P.U.'s contribution was supported by a National Geographic Waitt Grant (W421-16). X.J. and W.Z.'s contribution was supported by the Chinese Natural Science Foundation (41602019) and the State Key Laboratory of Palaeobiology and Stratigraphy (Nanjing Institute of Geology and Palaeontology, CAS [163120]).

Acknowledgements. P.D.M. and P.U. wish to express our gratitude to the late Junchang Lü for his help facilitating access to both *Dongyangosaurus* and *Jiangshanosaurus*. P.D.M. also thanks Rengjun Chen at the Dongyang Museum for providing brief access to *Dongyangosaurus*, as well as Alejandro Kramarz and Martin Ezcurra (MACN), Qiqing Pang (HBV) and Gabrielle Baglione (MHNH), for facilitating access to *Antarctosaurus*, *Huabeisaurus* and *Normanniasaurus*, respectively. Comments by Stephen Poropat and an anonymous reviewer improved this manuscript. We gratefully acknowledge the Willi Hennig Society, which has sponsored the development and free distribution of TNT.

# Appendix A. Revised scores and added characters

The following changes were made to existing character scores (C1–542) in the matrix of Mannion *et al.* [71]. In each case, the first number denotes the character and the number/symbol in parentheses denotes the new score:

*Alamosaurus*: 403 (0); 407 (1)

*Chubutisaurus*: 212 (0); 217 (1)

*Daxiatitan*: 350 (?)

*Dongbeititan*: 350 (?)

*Dongyangosaurus*: 23 (1); 25 (0); 26 (1); 33 (1); 145 (0/1); 148 (2); 153 (1); 165 (1); 197 (0); 203 (0); 245 (0); 247 (0); 252 (1); 253 (1); 279 (?); 333 (0); 340 (0); 342 (0); 345 (0); 349 (0); 352 (0); 354 (1); 380 (0); 383 (0); 384 (0); 385 (1); 387 (1); 419 (0); 473 (1); 475 (0); 479 (0); 481 (0); 482 (0); 497 (1); 498 (0); 499 (0); 522 (0); 525 (0); 526 (0); 527 (0); 530 (0); 532 (1)

*Isisaurus*: 98 (0); 99 (0)

*Jiangshanosaurus*: 23(1); 25 (1); 27 (0); 32 (?); 37 (?); 38 (?); 62 (0); 65 (1); 145 (1); 149 (1); 152 (0); 155 (0); 161 (?); 162 (1); 163 (0); 166 (0); 167 (1); 168 (0); 176 (0); 181 (0); 182 (0); 184 (?); 197 (?); 204 (0); 212 (0); 216 (0); 217 (1); 218 (?); 251 (0); 258 (0); 333 (0); 334 (1); 338 (1); 339 (?); 340 (0); 341 (1); 342 (0); 350 (1); 352 (0); 355 (1); 358 (0); 360 (0); 361 (0); 384 (0); 386 (0); 387 (0); 409 (1); 411 (1); 419 (0); 470 (0); 473 (0); 475 (0); 476 (1); 480 (0); 481 (0); 484 (0); 486 (1); 489 (0); 490 (0); 491 (0); 492 (1); 506 (0); 507 (0); 515 (?); 525 (0); 532 (1); 533 (1)

*Malawisaurus*: 99 (1); 315 (0); 316 (1); 318 (0); 439 (0); 440 (0)

*Ruyangosaurus*: 15 (1); 17 (0); 20 (0); 21 (1); 23 (1); 24 (1); 25 (1); 65 (1); 66 (1); 115 (2); 118 (1); 119 (0); 120 (1); 121 (0); 122 (0/1); 124 (0); 125 (0); 144 (0); 145 (1); 146 (1); 156 (1); 159 (0); 162 (1); 163 (0); 165 (0); 167 (1); 168 (0); 169 (0); 170 (1); 174 (1); 178 (0); 244 (1); 245 (0); 246 (1); 248 (0); 257 (0); 258 (0); 259 (2); 323 (0); 324 (0); 332 (0); 337 (1); 342 (?); 343 (0); 346 (1); 347 (0); 348 (0); 380 (0); 382 (0); 383 (0); 389 (1); 456 (0); 457 (0); 466 (0); 471 (0); 473 (1); 477 (0); 480 (0); 481 (0); 482 (?); 484 (0); 485 (1); 486 (1); 520 (0); 534 (0); 535 (0)

Six characters were added to the end of the character list, as C543–C548:

C543. Maxilla, tab-like process on the posterior surface of the post-dentigerous portion: absent (0); present (1) (new character: based on [32]; only scored for taxa with an emarginated ventral margin of the jugal process of the maxilla).

C544. Frontal, lateral margin: level with medial margin (0); raised relative to medial margin (1) (new character: based on [85]).

C545. Basal tubera, small ventrolateral process set off by a notch: absent (0); present (1) (new character: based on [85,87]).

C546. Teeth, rounded boss-like structures ('buttresses' or 'cingular cusps') on mesial and distal margins of lingual surface, close to the base of the crown: absent (0); present (1) (new character: based on [26,54,97,118]).

C547. Anterior-middle caudal centra, anterior margin relative to anteroposterior axis of centrum, in lateral view: perpendicular (0); inclined forward (faces anteroventrally) (1) [88].

C548. Fibula, anterolateral trochanter, situated on proximal third: absent (0); present (1) [132]; modified here based on: [85,133].

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
