## [Reviewer comments · Royal Society Open Science]

Review History

RSOS-191057.R0 (Original submission)

Review form: Reviewer 1 (Stephen F. Poropat)

Is the manuscript scientifically sound in its present form?

Yes

Are the interpretations and conclusions justified by the results?

Yes

Is the language acceptable?

Yes

Do you have any ethical concerns with this paper?

No

Have you any concerns about statistical analyses in this paper?

No

Recommendation?

Accept as is

Comments to the Author(s)

Apart from a few very minor typographic changes, and the amendment of the PMU specimen's number, the manuscript is ready for publication as is. Would have been great to have some images of Dongyangosaurus but it would seem that this was not possible. Also, interesting to note how different the tree topology was from other iterations of this matrix...

Review form: Reviewer 2 (Jose Luis Carballido)

Is the manuscript scientifically sound in its present form?

Yes

Are the interpretations and conclusions justified by the results?

Yes

Is the language acceptable?

Yes

Do you have any ethical concerns with this paper?

No

Have you any concerns about statistical analyses in this paper?

Yes

Recommendation?

Accept with minor revision (please list in comments)

Comments to the Author(s)

The manuscript presented by Mannion and collaborators re-describes in detail the materials from Jiangshanosaurus, at the time that they presented new information from another important taxon from China (Dongyangosaurus). Besides the further anatomical details coming from the descriptions, the manuscript results of great importance and impact due to the low information available from Asiatic somphospondylans and how they could impact on our current knowledge about the evolution of this clade. I found the manuscript interesting as many of the Laurasian taxa introduced in the analysis are generally not considered in discussion on evolution and paleobiogeography. The authors call the attention on that. So, I think this manuscript will be helpful to understand the evolution of somphospondylan sauropods further.

The descriptions are detailed, precise, and well figured. My primary comment is about the use of two different criteria for running the analysis (under equal weights and under implied weighting). I am in agree in using different approaches for the tree searches, other than the conventional parsimony analysis under equally weighting characters, but I think that a short mention of what implied weighting analysis means will help readers to understand why the authors decide to try these two methods.

Additionally, and most important, is the decision on the K used for the implied weight analysis.

By default, the TNT uses a K=3. Nevertheless, higher values are recommended to be used (as Goloboff et al. noted in 2008 and more recently in Goloboff 2017; see references below). So, I do not think that the use of the default K will be the better choice for the implied weighting analysis, and I will recommend the use of a higher value (8-12 sounds more accurate).

In page 15 (lines 10-17), the authors mention that Jiangshanosaurus present many plesiomorphic features for Titanosauriformes. Based on most previous analyses, I agree with that. But, what about the EIW analysis in which Andesaurus is recovered in a basal position, and therefore most somphospondylans are titanosaurs?. The same for the following paragraph and the position of Andesaurus. This discussion fit perfectly with the topology obtained in the most previous analysis but not for the EIW results.

I think, that if the authors decide to re-run the EIW analysis and they find similar results to that showed here, a short comment should be introduced to clarify the use of systematic statements. For example, I like the idea of referring some taxa as basal somphospondylans if, irrespective of the position of Andesaurus, they are recovered in both analyses closer to the Titanosauriformes node than to Saltasaurus (as discussed for Tastavinsaurus and Europatitan). But, sounds some confusing when Titanosauria is used, as based on EWP and EIW the node that defines Titanosauria changes drastically (e.g., page 18, line 39; Our new topologies also suggest that Baotianmansaurus henanensis, Huanghetitan liujiaxiaensis and 'Huanghetitan' ruyangensis probably lie outside of Titanosauria too). Here will be better to mention that these taxa are basal somphospondylans instead of non-titanosaurs, as they are titanosaurs in the EIW analysis.

Besides these comments I found the paper very interesting, and I think it is a significant contribution to alert further researchers into including Laurasian taxa in phylogenetic and/or paleo-biogeographic discussion.

Goloboff, P., Carpenter, J., Arias, J., Miranda-Esquivel, D., 2008. Weighting against homoplasy improves phylogenetic analysis of morphological data sets. *Cladistics* 24, 1-16.

Goloboff, P., Torres, A., Arias, S. J. 2017. Weighted parsimony outperforms other methods of phylogenetic inference under models appropriate for morphology. *Cladistics* 34, 407-437.

Decision letter (RSOS-191057.R0)

26-Jul-2019

Dear Dr Mannion

On behalf of the Editors, I am pleased to inform you that your Manuscript RSOS-191057 entitled "New information on the Cretaceous sauropod dinosaurs of Zhejiang Province, China: impact on Laurasian titanosauriform phylogeny and biogeography" has been accepted for publication in Royal Society Open Science subject to minor revision in accordance with the referee suggestions. Please find the referees' comments at the end of this email. Note that the most important matters to address are the few typographic changes, and the correction of a specimen number. These changes will not take very long to complete.

The reviewers and handling editors have recommended publication, but also suggest some minor revisions to your manuscript. Therefore, I invite you to respond to the comments and revise your manuscript.

- Ethics statement

- Data accessibility

If you wish to submit your supporting data or code to Dryad (<http://datadryad.org/>), or modify your current submission to dryad, please use the following link:
<http://datadryad.org/submit?journalID=RSOS&manu=RSOS-191057>

- Competing interests

- Authors' contributions

- Acknowledgements

- Funding statement

Please ensure you have prepared your revision in accordance with the guidance at <https://royalsociety.org/journals/authors/author-guidelines/> -- please note that we cannot publish your manuscript without the end statements. We have included a screenshot example of

the end statements for reference. If you feel that a given heading is not relevant to your paper, please nevertheless include the heading and explicitly state that it is not relevant to your work.

Because the schedule for publication is very tight, it is a condition of publication that you submit the revised version of your manuscript before 04-Aug-2019. Please note that the revision deadline will expire at 00.00am on this date. If you do not think you will be able to meet this date please let me know immediately.

Please note that Royal Society Open Science charge article processing charges for all new submissions that are accepted for publication. Charges will also apply to papers transferred to Royal Society Open Science from other Royal Society Publishing journals, as well as papers

submitted as part of our collaboration with the Royal Society of Chemistry (<http://rsos.royalsocietypublishing.org/chemistry>).

on behalf of Dr Julia Brenda Desojo (Associate Editor) and Jon Blundy (Subject Editor)
openscience@royalsociety.org

Reviewer comments to Author:
Reviewer: 1

Comments to the Author(s)

Apart from a few very minor typographic changes, and the amendment of the PMU specimen's number, the manuscript is ready for publication as is. Would have been great to have some images of *Dongyangosaurus* but it would seem that this was not possible. Also, interesting to note how different the tree topology was from other iterations of this matrix...

Reviewer: 2

Comments to the Author(s)

The manuscript presented by Mannion and collaborators re-describes in detail the materials from *Jiangshanosaurus*, at the time that they presented new information from another important taxon from China (*Dongyangosaurus*). Besides the further anatomical details coming from the descriptions, the manuscript results of great importance and impact due to the low information available from Asiatic somphospondylans and how they could impact on our current knowledge about the evolution of this clade. I found the manuscript interesting as many of the Laurasian taxa introduced in the analysis are generally not considered in discussion on evolution and paleobiogeography. The authors call the attention on that. So, I think this manuscript will be helpful to understand the evolution of somphospondylan sauropods further.

The descriptions are detailed, precise, and well figured. My primary comment is about the use of two different criteria for running the analysis (under equal weights and under implied weighting). I am in agree in using different approaches for the tree searches, other than the conventional parsimony analysis under equally weighting characters, but I think that a short mention of what implied weighting analysis means will help readers to understand why the authors decide to try these two methods.

Additionally, and most important, is the decision on the K used for the implied weight analysis. By default, the TNT uses a K=3. Nevertheless, higher values are recommended to be used (as

Goloboff et al. noted in 2008 and more recently in Goloboff 2017; see references below). So, I do not think that the use of the default K will be the better choice for the implied weighting analysis, and I will recommend the use of a higher value (8-12 sounds more accurate).

In page 15 (lines 10-17), the authors mention that Jiangshanosaurus present many plesiomorphic features for Titanosauriformes. Based on most previous analyses, I agree with that. But, what about the EIW analysis in which Andesaurus is recovered in a basal position, and therefore most somphospondylans are titanosaurs?. The same for the following paragraph and the position of Andesaurus. This discussion fit perfectly with the topology obtained in the most previous analysis but not for the EIW results.

I think, that if the authors decide to re-run the EIW analysis and they find similar results to that showed here, a short comment should be introduced to clarify the use of systematic statements. For example, I like the idea of referring some taxa as basal somphospondylans if, irrespective of the position of Andesaurus, they are recovered in both analyses closer to the Titanosauriformes node than to Saltasaurus (as discussed for Tastavinsaurus and Europatitan). But, sounds some confusing when Titanosauria is used, as based on EWP and EIW the node that defines Titanosauria changes drastically (e.g., page 18, line 39; Our new topologies also suggest that Baotianmansaurus henanensis, Huanghetitan liujiaxiaensis and 'Huanghetitan' ruyangensis probably lie outside of Titanosauria too). Here will be better to mention that these taxa are basal somphospondylans instead of non-titanosaurs, as they are titanosaurs in the EIW analysis.

Besides these comments I found the paper very interesting, and I think it is a significant contribution to alert further researchers into including Laurasian taxa in phylogenetic and/or paleo-biogeographic discussion.

Goloboff, P., Carpenter, J., Arias, J., Miranda-Esquivel, D., 2008. Weighting against homoplasy improves phylogenetic analysis of morphological data sets. *Cladistics* 24, 1-16.

Goloboff, P., Torres, A., Arias, S. J. 2017. Weighted parsimony outperforms other methods of phylogenetic inference under models appropriate for morphology. *Cladistics* 34, 407-437.

Author's Response to Decision Letter for (RSOS-191057.R0)

See Appendix A.

Decision letter (RSOS-191057.R1)

01-Aug-2019

Dear Dr Mannion,

I am pleased to inform you that your manuscript entitled "New information on the Cretaceous sauropod dinosaurs of Zhejiang Province, China: impact on Laurasian titanosauriform phylogeny and biogeography" is now accepted for publication in Royal Society Open Science.

on behalf of Dr Julia Brenda Desojo (Associate Editor) and Jon Blundy (Subject Editor)
openscience@royalsociety.org

Follow Royal Society Publishing on Twitter: [@RSocPublishing](https://twitter.com/RSocPublishing)
Follow Royal Society Publishing on Facebook:
<https://www.facebook.com/RoyalSocietyPublishing.FanPage/>
Read Royal Society Publishing's blog: <https://blogs.royalsociety.org/publishing/>

Appendix A

Reviewer comments to Author:

Reviewer: 1

Comments to the Author(s)

Apart from a few very minor typographic changes, and the amendment of the PMU specimen's number, the manuscript is ready for publication as is. Would have been great to have some images of *Dongyangosaurus* but it would seem that this was not possible. Also, interesting to note how different the tree topology was from other iterations of this matrix...

Response: We have corrected the typographic changes highlighted by the reviewer (with the exception of retaining 'predating' rather than 'pre-dating'), as well as the amended specimen number. We agree it would be great to include photographs of the *Dongyangosaurus* material, but these were all taken through glass, in poor lighting and, by necessity, at unorthodox angles, therefore greatly limiting their utility.

Reviewer: 2

Comments to the Author(s)

The manuscript presented by Mannion and collaborators re-describes in detail the materials from *Jiangshanosaurus*, at the time that they presented new information from another important taxon from China (*Dongyangosaurus*). Besides the further anatomical details coming from the descriptions, the manuscript results of great importance and impact due to the low information available from Asiatic somphospondylans and how they could impact on our current knowledge about the evolution of this clade. I found the manuscript interesting as many of the Laurasian taxa introduced in the analysis are generally not considered in discussion on evolution and paleo-biogeography. The authors call the attention on that. So, I think this manuscript will be helpful to understand the evolution of somphospondylan sauropods further.

Response: We thank the reviewer for their positive comments. No actions requested.

The descriptions are detailed, precise, and well figured. My primary comment is about the use of two different criteria for running the analysis (under equal weights and under implied weighting). I am in agree in using different approaches for the tree searches, other than the conventional parsimony analysis under equally weighting characters, but I think that a short mention of what implied weighting analysis means will help readers to understand why the authors decide to try these two methods. Additionally, and most important, is the decision on the K used for the implied weight analysis. By default, the TNT uses a K=3. Nevertheless, higher values are recommended to be used (as Goloboff et al. noted in 2008 and more recently in Goloboff 2017; see references below). So, I do not think that the use of the default K will be the better choice for the implied weighting analysis, and I will recommend the use of a higher value (8-12 sounds more accurate).

Response: The reviewer makes an excellent point. Rather than replacing our analysis using a k-value of 3, we have instead added a third analysis, using a k-value of 9. We have also provided some explanation of what extended implied weighting actually is, including incorporating the suggested Goloboff reference.

In page 15 (lines 10-17), the authors mention that Jiangshanosaurus present many plesiomorphic features for Titanosauriformes. Based on most previous analyses, I agree with that. But, what about the EIW analysis in which Andesaurus is recovered in a basal position, and therefore most somphospondylans are titanosaurs?. The same for the following paragraph and the position of Andesaurus. This discussion fit perfectly with the topology obtained in the most previous analysis but not for the EIW results. I think, that if the authors decide to re-run the EIW analysis and they find similar results to that showed here, a short comment should be introduced to clarify the use of systematic statements. For example, I like the idea of referring some taxa as basal somphospondylans if, irrespective of the position of Andesaurus, they are recovered in both analyses closer to the Titanosauriformes node than to Saltasaurus (as discussed for Tastavinsaurus and Europatitan). But, sounds some confusing when Titanosauria is used, as based on EWP and EIW the node that defines Titanosauria changes drastically (e.g., page 18, line 39; Our new topologies also suggest that Baotianmansaurus henanensis, Huanghetitan liujiaxiaensis and 'Huanghetitan' ruyangensis probably lie outside of Titanosauria too). Here will be better to mention that these taxa are basal somphospondylans instead of non-titanosaurs, as they are titanosaurs in the EIW analysis.

Response: We have tried to clarify this issue throughout the text. Early in our Discussion we have added "Below, our use of the clade name Titanosauria refers to the topology recovered in our EWP analysis" and, when appropriate, we have used terms like 'basal somphospondylan' rather than non-titanosaur. We had already noted that this 'stemward slippage' of Andesaurus has occurred in all previous iterations of this matrix (going back to 2013), and so it is not a novel result.